# Investigation of Co–Fe–Al Catalysts for High-Calorific Synthetic Natural Gas Production: Pilot-Scale Synthesis of Catalysts

**Tae Young Kim** [1] , **Seong Bin Jo** [2,3] , **Jin Hyeok Woo** [1] , **Jong Heon Lee** [1] , **Ragupathy Dhanusuraman** [4] , **Soo Chool Lee** [3,*] **and Jae Chang Kim** [1,*]

[1] Department of Chemical Engineering, Kyungpook National University, Daegu 41566, Korea; tyoung0218@knu.ac.kr (T.Y.K.); wjh8865@knu.ac.kr (J.H.W.); rnswma123@knu.ac.kr (J.H.L.)
[2] Department of Chemical and Environmental Engineering, University of California–Riverside, Riverside, CA 92521, USA; sjo016@ucr.edu
[3] Research Institute of Advanced Energy Technology, Kyungpook National University, Daegu 41566, Korea
[4] Department of Chemistry, National Institute of Technology Puducherry, Karaikal 609609, India; ragu.nitpy@gmail.com
[*] Correspondence: soochool@knu.ac.kr (S.C.L.); kjchang@knu.ac.kr (J.C.K.); Tel.: +82-53-950-5622 (S.C.L. & J.C.K.)

**Abstract:** Co–Fe–Al catalysts prepared using coprecipitation at laboratory scale were investigated and extended to pilot scale for high-calorific synthetic natural gas. The Co–Fe–Al catalysts with different metal loadings were analyzed using BET, XRD, $H_2$-TPR, and FT-IR. An increase in the metal loading of the Co–Fe–Al catalysts showed low spinel phase ratio, leading to an improvement in reducibility. Among the catalysts, 40CFAl catalyst prepared at laboratory scale afforded the highest $C_2$–$C_4$ hydrocarbon time yield, and this catalyst was successfully reproduced at the pilot scale. The pelletized catalyst prepared at pilot scale showed high CO conversion (87.6%), high light hydrocarbon selectivity ($CH_4$ 59.3% and $C_2$–$C_4$ 18.8%), and low byproduct amounts ($C_{5+}$: 4.1% and $CO_2$: 17.8%) under optimum conditions (space velocity: 4000 mL/g/h, 350 °C, and 20 bar).

**Keywords:** high-calorific synthetic natural gas (HC-SNG); cobalt–iron–alumina; coprecipitation; loading amount; pilot-scale synthesis

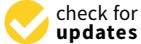



## 1. Introduction

Synthetic natural gas (SNG; CO + 3$H_2$ = $CH_4$ + $H_2O$), produced from coal and biomass, has received considerable attention as a substitute for fossil fuels, because it mainly consists of $CH_4$, which emits the smallest amount of $CO_2$ per energy unit among fossil fuels [1–5]. However, the heating value of SNG (9520 kcal/$Nm^3$) is lower than the standard heating value (10,400 kcal/$Nm^3$) for power generation in South Korea and Japan [6–14]. Consequently, liquefied petroleum gas (LPG; $C_3$ and $C_4$) must be added to increase the heating value of SNG. This process is vulnerable to price fluctuations because the price of LPG strongly depends on oil prices. To overcome the problems arising from the relatively low heating value, several researchers have proposed "high-calorific synthetic natural gas (HC-SNG)" obtained by producing $C_2$–$C_4$ as well as $CH_4$ using the Fischer–Tropsch (FT) reaction [6–14]. Among transition metal catalysts, Ni is employed for methanation applications because of its high catalytic activity, high $CH_4$ selectivity, and comparatively low cost [3–5,11]. Thus, Ni-based catalysts exhibit low $C_{2+}$ selectivity. Inui et al. published the first study on the synthesis of HC-SNG from a coke oven gas using a Co-based catalyst [6]. This process achieved an enhanced heating value (10,020 kcal/$Nm^3$) using synthetic light hydrocarbons ($C_2$–$C_4$) during the SNG process, without LPG addition. However, Co-based catalysts often exhibit low $C_2$–$C_4$ selectivity. Lee et al. examined the activity of Fe-based catalysts for achieving higher $C_2$–$C_4$ hydrocarbon selectivities compared to those of Co-based catalysts. Carburized Fe-based catalysts achieved a high CO

conversion and $C_2$–$C_4$ selectivity because of the enhanced CO adsorption on Fe carbides and the relatively large Brunauer–Emmett–Teller (BET) surface area of the carburized catalysts [8]. In an earlier report, we studied the activity and selectivity of Co–Fe bimetallic catalysts for the production of HC-SNG [11]. The catalysts with a high Co/Fe ratio showed high methane and low $C_2$–$C_4$ selectivities, while low methane and high $C_2$–$C_4$ selectivities were obtained at a low Co/Fe ratio. The bimetallic Co–Fe/$Al_2O_3$ catalyst with a Co:Fe ratio of 1:3 displayed the highest $C_2$–$C_4$ selectivity (28.2%) at a high CO conversion (91.5%) because the presence of Co enhanced the reducibility of Fe. Moreover, the effects of the $H_2$/CO gas ratio and the reaction temperature on the catalytic performance were investigated. The optimum conditions for the production of $C_2$–$C_4$ hydrocarbons were found to be as follows: $H_2$/CO ratio = 3.0, reaction temperature = 300 °C, and reaction pressure = 10 bar.

In this study, we prepared Co–Fe–Al catalysts using a coprecipitation method, to exploit its component distribution homogeneity, high reproducibility, and economic advantage for industrial syntheses [15,16]. The objective of the present work is to study the catalytic performance with different metal (Co and Fe) loadings on a coprecipitated catalyst for pilot-scale synthesis. In addition, the catalytic performance of a catalyst prepared at a pilot scale was investigated. The catalysts were characterized by inductively coupled plasma optical emission spectrometry (ICP-OES), Brunauer–Emmett–Teller (BET) analysis, X-ray diffraction (XRD), Fourier-transform infrared spectroscopy (FT-IR), and $H_2$-temperature-programmed reduction (TPR).

## 2. Results and Discussion

### 2.1. Characteristics of the Co–Fe–Al Catalysts

The textural properties and ICP-OES of the CFAl catalyst with different metal loadings are listed in Table 1. The contents of Co and Fe in the coprecipitated catalyst achieved the designed value (Co/Fe ratio = 1/3). The BET surface area increased from 183 to 241.5 $m^2$/g by increasing the Co and Fe loadings from 20 to 40 wt.%, and a further increase in the Co and Fe loadings caused a decrease in the BET surface area (214.9 $m^2$/g). The pore size distribution of the 20CFAl catalyst showed a mesoporous structure with diameters between 10 and 40 nm (Figure S1). With an increasing metal loading amount, an additional pore structure (4–8 nm) was observed, leading to an increase in the BET surface area. The BET surface area increased with the increasing metal atomic ratio when the metal/Al atomic ratio was lower than the stoichiometric spinel ratio (33.3%) because the spinel phase contains a greater mesoporous structure compared to the alumina phase [17]. As shown in Table 1, the atomic ratio of CFAl catalysts approached to the stoichiometry ratio of spinel phase with increasing metal loading. This phenomenon might have led to the increase in the BET surface area and the formation of a mesoporous structure. However, the decrease in the BET surface area of the 50CFAl catalyst was due to the crystallization of metal oxides (Co and Fe) [17–19].

**Table 1.** Characterization of the CFAl catalysts with different metal loadings.

| Notation [a] | Metal Content (wt.%) [b] | | Relative Atomic Ratio of Metal (%) | BET Surface Area ($m^2$/g) | Crystalline Size (nm) | | |
|---|---|---|---|---|---|---|---|
| | Co | Fe | | | Spinel [c] | $Fe_3O_4$ [d] | $Fe^0$ [d] |
| 20CFAl | 4.5 | 13.6 | 9.5 | 183.6 | 3.4 | - | - |
| 30CFAl | 6.6 | 22.3 | 16.2 | 201.9 | 3.5 | - | 4.4 |
| 40CFAl | 10.1 | 29.9 | 24.0 | 241.5 | 4.0 | 4.0 | 10.2 |
| 50CFAl | 12.8 | 37.4 | 32.4 | 214.9 | 4.1 | 12.6 | 7.2 |

[a] The catalyst was denoted as xCFAl for Co–Fe–Al with different metal (Co and Fe) concentrations of x, from 20 to 50 wt.% with $Al_2O_3$ support and a fixed Co/Fe ratio of 1/3. [b] Metal content was determined by ICP-OES. [c] Determined from Scherrer's equation from the calcined catalyst in XRD patterns. [d] Determined from Scherrer's equation from the reduced catalyst in XRD patterns.

The XRD patterns of CFAl catalysts in the calcined and reduced states are shown in Figure 1. The XRD patterns of the calcined CFAl catalysts do not exhibit Co oxide, Fe oxide, and alumina. Interestingly, only the spinel-like phase with characteristic diffraction peaks at 2θ values of 31°, 37°, 45°, and 65° was observed in all the catalysts with different metal loadings [20–23]. The spinel-like structures correspond to the phases of $CoAl_2O_4$ (01-082-2245 JCPDS) and $FeAl_2O_4$ (00-034-0192 JCPDS). However, distinguishing $CoAl_2O_4$ from $FeAl_2O_4$ is difficult due to the very close resemblance in diffraction patterns. This is because the Co, Fe, and Al species interacted intimately with each other, resulting in the formation of a spinel phase in the CFAl catalysts during the calcination processes. The calcined CFAl catalysts showed broad peaks, indicating that all the catalysts consist of well dispersed amorphous phases. Moreover, the calculated crystallite size of spinel phase in all catalysts was similar but increased slightly with the increasing metal loading amount, as shown in Table 1. In the case of the reduced state, the 20CFAl catalyst showed no significant change in peaks after reduction, suggesting that metal species were probably not reduced. In contrast, with increasing metal loading, the additional diffraction peaks of $CoFe_2O_4$, $Fe_3O_4$, and $Fe^0$ were observed. For the 30CFAl catalysts, spinel-like structures existed even after reduction, but the peaks of $Fe^0$ were observed compared to the 20CFAl. In particular, the diffraction peaks $Fe^0$ were clearly observed for the 40CFAl and 50CFAl catalysts, thereby suggesting that an increase in metal loading might enhance the reducibility. However, as shown in Table 1, the crystallite size of the $Fe_3O_4$ phase of 50CFAl was larger than that of 40CFAl. It has been reported that $Fe_2O_3$ gradually reduces into $Fe^0$ in situ XRD during reduction [24]. Thus, for the 50CFAl catalyst, the higher peaks of $Fe_2O_3$ might have been due to the limited time for the reduction of Fe oxide phase in the present study.

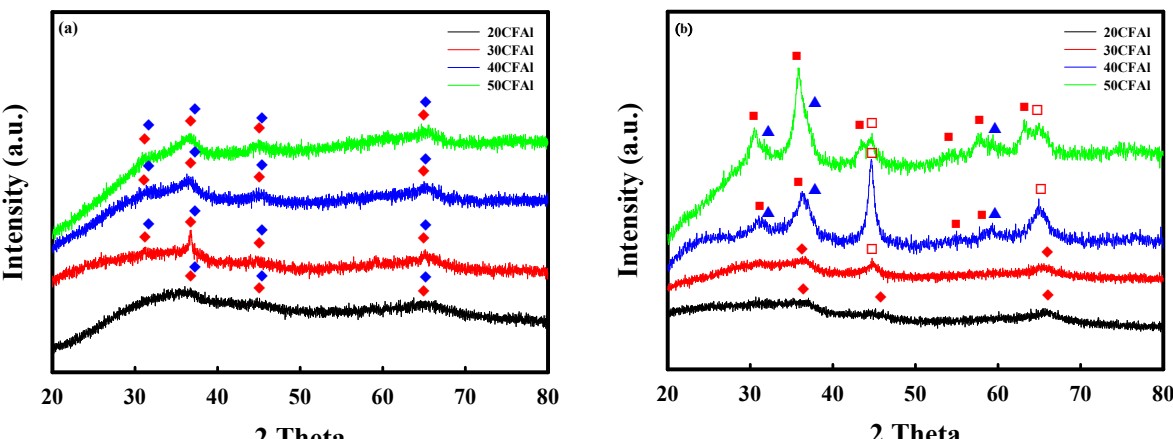

**Figure 1.** XRD patterns of the calcined and reduced CFAl: (**a**) calcined and (**b**) $H_2$-reduced at 500 °C; (◆) $FeAl_2O_4$, (◆) $CoAl_2O_4$, (▲) $CoFe_2O_4$, (■) $Fe_3O_4$, and (□) Fe metal.

$H_2$-TPR profiles of the CFAl catalysts with different metal loadings are given in Figure 2. The $H_2$-TPR curve of all the CFAl catalysts exhibits two reduction peaks. The low-temperature peak (at 333–369 °C) is associated with the reductions of $Co_3O_4$ to CoO and $Fe_2O_3$ to $Fe_3O_4$ [11,20]. The high-temperature (at 387–705 °C) peak may be ascribed to the sequential reduction of CoO to Co, $Fe_3O_4$ to Fe, and the spinel-like structure ($CoAl_2O_4$ and $FeAl_2O_4$) above 700 °C [11,20,23]. For the low-temperature peak, no significant change was identified with an increasing metal loading. Conversely, the high-temperature peak shifted drastically toward relatively low temperatures and exhibited increased sharpness with increasing metal loading. Thus, it could be concluded that CFAl catalysts were more easily reduced at high metal loadings.

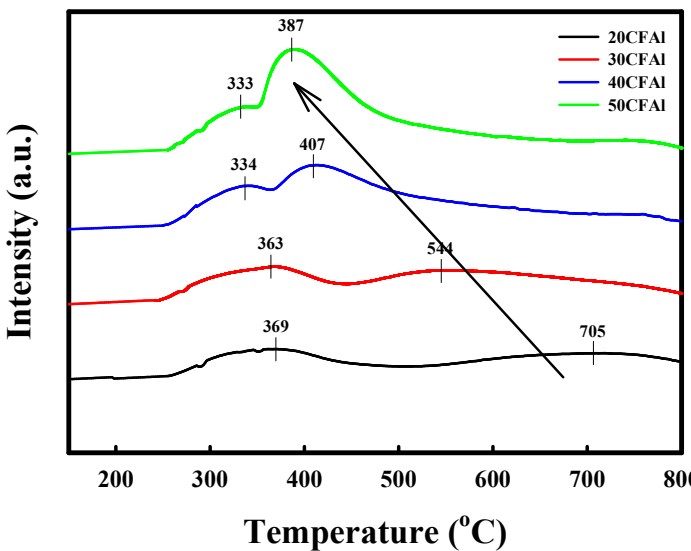

**Figure 2.** $H_2$-TPR profiles of CFAl with different metal loadings.

The FT-IR spectra of the CFAl catalysts in the calcined and reduced states were analyzed to determine the reason for the enhanced reducibility of the CFAl catalysts at high metal loading amounts. The IR spectra of the calcined CFAl catalysts contain an absorption band at 3390 $cm^{-1}$, indicating the $-OH$ stretching vibrations. The absorption band at ~1618 $cm^{-1}$ corresponds to the vibration of $H_2O$ molecules. The transmission bands at 1361 and 1483 $cm^{-1}$ are due to the carbonyl group of the carboxylate ions, which may remain adsorbed on the surface of $Al_2O_3$ during precipitation. The peaks between 778 and 850 $cm^{-1}$ are associated with the metal–Al spinel. This result agrees with the result of the XRD analysis. Contrary to the results of the XRD analysis, the metal–oxygen vibration was observed at 400–600 $cm^{-1}$ in addition to the spinel phase. With an increasing metal loading, the band intensity of the metal–Al spinel sharply reduced, because a relatively high metal content leads to a low ratio of the spinel-like phase [21,23]. Contrarily, the adsorption bands at 400–600 $cm^{-1}$, corresponding to the metal–oxygen vibrations, did not change. Figure 3b shows the FT-IR spectra of 20CFAl and 40CFAl catalysts after reduction at 500 °C; the results of the catalysts in calcined states are depicted for comparison. As shown in Figure 3b, the metal–Al spinel changed rarely at both the 20CFAl and 40CFAl catalysts after reduction at 500 °C. For the 40CFAl catalyst, however, the adsorption bands of metal–oxygen decreased considerably compared to the 20CFAl catalyst, indicating that the metal oxide was easily reduced. At a low metal content, the catalysts mainly comprised a spinel form, which was not easy to reduce, as mentioned above. Conversely, the catalysts comprised the high ratio of metal oxides (Co or Fe) to spinel phase. Therefore, it can be concluded that the enhanced reducibility of CFAl catalysts at high metal loading was due to the high ratio of metal oxides to the spinel phase.

### 2.2. Catalytic Performance of the Co–Fe–Al Catalysts

In our previous papers, bimetallic Co–Fe catalysts with a Co:Fe ratio of 1:3 exhibited a high yield toward the light hydrocarbons ($C_2$–$C_4$) in SNG ($H_2$/CO = 3.0) compared to Co catalysts because the presence of Co enhanced the reducibility of Fe [11,14]. The catalytic performance of CFAl catalysts developed in the present study as the function of metal loading are shown in Figure 4 and summarized in Table 2. No remarkable change was observed in CO conversion and selectivity for 10 h (Figure S2). With the increasing metal loading, the initial CO conversion increased remarkably as follows: 20CFAl (17.8%) < 30CFAl (54.4%) < 40CFAl (90.2%) < 50CFAl (97.0%). The fact that CO conversion increased dramatically with an increasing metal loading might have been due to the enhanced reducibility from the low ratio of spinel phases at high metal loadings compared to the 20CFAl catalyst. It is known that the metallic phase can provide reaction

sites for CO hydrogenation [20,22]. However, above the 40CFAl catalyst, no significant change in the CO conversion was observed. These results correspond to the characteristic analysis including XRD, $H_2$-TPR, and FT-IR. The CFAl catalysts in the present study showed the similar trend of hydrocarbon selectivity with the values of the bimetallic Co–Fe catalysts with a Co:Fe ratio of 1:3 as in previous studies [11]. The $CH_4$ selectivity decreased from 39.4% to 27.7%, the $C_2$–$C_4$ selectivity increased from 28.5% to 30.7%, and the $C_{5+}$ selectivity increased from 7.6% to 16.4% with increasing metal loading, except for the 20CFAl catalyst. This is because the reducibility of the Fe oxide phase increased with increasing metal loading, as mentioned above. It is well known that the $CH_4$ selectivity of Fe catalysts decreases with increasing CO conversion at high pressures due to the enhanced readsorption and reinsertion of olefins [11]. The $CO_2$ selectivity increased linearly with metal loading from 20 to 50wt.%. This increase in $CO_2$ selectivity was caused by the (WGS) reaction ($CO + H_2O = CO_2 + H_2$) at high $H_2O$ partial pressures; water vapor was produced in CO hydrogenation (paraffins: $nCO + (2n + 1)H_2 = C_nH_{2n+2} + nH_2O$, and olefins: $nCO + 2nH_2 = C_nH_{2n} + nH_2O$) [11,13,25]. This WGS reaction also leads to an increase in $H_2$/CO ratio, which affects the hydrocarbon selectivity in a Fischer–Tropsch reaction [13]. In the case of the 20CFAl catalyst, the $H_2$/CO ratio adjustment is small due to the lower CO conversion compared to other catalysts [13]. Thus, comparing the hydrocarbon selectivity of 20CFAl with that of other catalysts is difficult. Furthermore, the paraffin ratio (P/(P + O)) was calculated with the increasing metal amount, where P and O represent the paraffins and olefins in the $C_2$–$C_4$, respectively. The paraffin ratio also increased from 0.60 to 0.94 with increasing CO conversion. The high $H_2$/CO ratio from WGS increases with increasing CO conversion, resulting in the improvement of the paraffin ratio [13]. In addition, it was found that the $C_2$–$C_4$ hydrocarbon time yield increased with increasing metal loading below 40 wt.%. The 40CFAl catalyst displayed the highest hydrocarbon time yield of 2.49 $mmol_{CO} \cdot g_{metal}^{-1} \cdot h^{-1}$. With a further increasing of the metal loading, the hydrocarbon time yield of 50CFAl was reduced to 2.24 $mmol_{CO} \cdot g_{metal}^{-1} \cdot h^{-1}$. Contrary to the increase in the metal loading amount (above 40 wt.%), no significant change in $C_2$–$C_4$ hydrocarbon yield was observed. Thus, the hydrocarbon time yield of 40CFAl was higher than that of 50CFAl. Of all the catalysts, 40CFAl was chosen as the HC-SNG catalyst for the following pilot-scale synthesis of the catalyst because it has the highest $C_2$–$C_4$ hydrocarbon time yield despite its lower CO conversion compared to the 50CFAl catalyst.

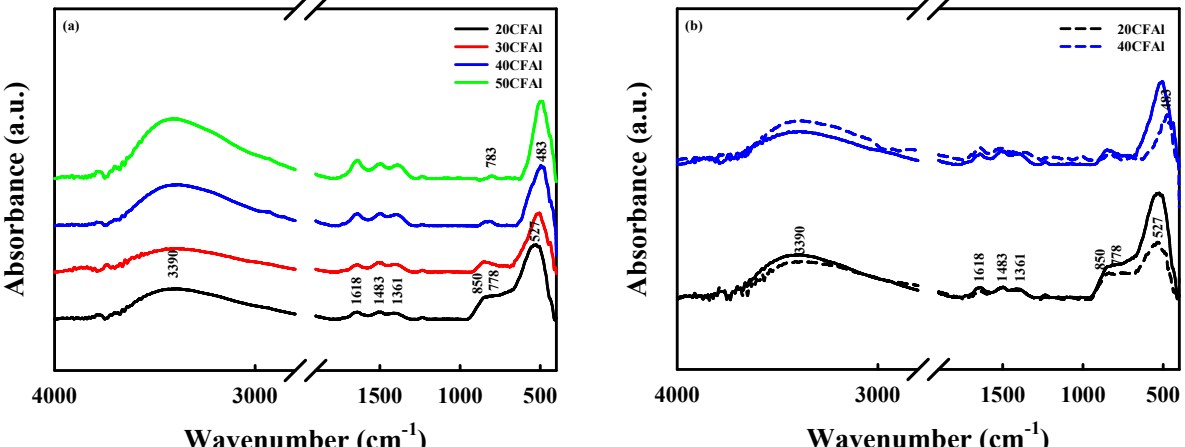

**Figure 3.** FT-IR spectra of the calcined and reduced CFAl catalysts: (**a**) calcined and (**b**) $H_2$-reduced at 500 °C. In frame (**b**), the calcined samples are depicted for comparison.

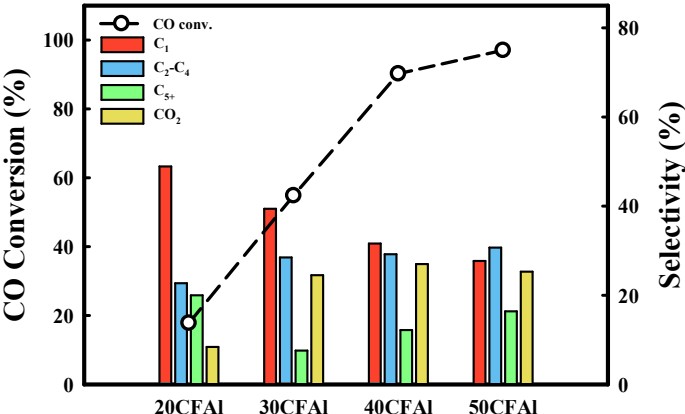

**Figure 4.** CO conversion and product selectivity of the CFAl catalysts.

**Table 2.** Summary of the catalytic performances of the CFAl Catalysts.

| Notation | CO Conversion (%) | Selectivity (%) | | | | P/(P + O) | $C_2-C_4$ Hydrocarbon Time Yield $(mmol_{CO} \cdot g_{metal}^{-1} \cdot h^{-1})$ [a] |
|---|---|---|---|---|---|---|---|
| | | CH$_4$ | $C_2-C_4$ | $C_{5+}$ | $CO_2$ | | |
| 20CFAl | 17.8 | 48.9 | 22.7 | 20.0 | 8.4 | 0.60 | 0.84 |
| 30CFAl | 54.4 | 39.4 | 28.5 | 7.6 | 22.1 | 0.85 | 2.04 |
| 40CFAl | 90.2 | 31.6 | 29.2 | 12.2 | 27.0 | 0.94 | 2.49 |
| 50CFAl | 97.0 | 27.7 | 30.7 | 16.4 | 26.7 | 0.94 | 2.24 |

[a] The hydrocarbon time yield was the number of CO moles converted to $C_2-C_4$ hydrocarbons per gram of metal (Co and Fe) per hour.

### 2.3. Performance of Pelletized 40CFAl Catalysts

The scheme of the pilot-scale synthesis is presented in Figure 5a. The 40CFAl catalyst was chosen as the HC-SNG catalyst based on its characteristics and catalytic performance. As shown in Figure S3, the catalytic performance of the catalyst prepared at the pilot scale was investigated for comparison with the catalysts prepared on the laboratory scale. The catalytic performances of the laboratory- and pilot-scale samples were similar, indicating that the pilot-scale catalyst was successfully reproduced. Thereafter, the catalyst prepared at the pilot scale was pressed using an extruder to produce the pellet (5 × 5 mm²), denoted as 40CFAl_P. The morphology and distribution of the metal for 40CFAl_P were determined by SEM–EDS analysis, as shown in Figure 5b. EDS analysis confirmed the presence of Co, Fe, and Al, which validated the uniform distribution of the metal content within the composites. The catalytic performance of 40CFAl_P was investigated using a pellet type under the same conditions as those for 40CFAl. The CO conversion of 40CFAl_P was lower than that of 40CFAl. The lower CO conversion of 40CFAl_P than that of 40CFAl might be due to the mass-transfer limitation, resulting in strong concentration gradients that affect negatively catalytic activity [26–28]. Moreover, the 40CFAl_P catalyst showed a different hydrocarbon distribution from that of 40CFAl. The CH$_4$ selectivity was similar for the 40CFAl_P and 40CFAl catalysts, whereas the 40CFAl_P catalyst shows a higher selectivity for long-chain hydrocarbon. It has been reported that the long-chain hydrocarbons in Fischer–Tropsch synthesis are strongly related to the internal mass-transfer limitation within the pelletized catalyst. The light hydrocarbon was unfavorable to the pelletized catalyst by diffusion-limited α-olefins [26]. In an earlier report, we studied operating parameters such as space velocity, reaction pressure, and temperature, and the bimetallic Co–Fe catalysts showed a high $C_2$-$C_4$ yield with a high paraffin ratio under two conditions ((I) 6000 mL/g/h, 300 °C, and 10 bar; (II) 4000 mL/g/h, 350 °C, and 20 bar) [13]. Thus, the catalytic test of the 40CFAl_P catalyst was carried out under these two different conditions (I) and (II) to achieve the high yield of $C_2$-$C_4$ hydrocarbons. Figure 6a shows the CO conversion of the 40CFAl_P catalyst under a different condition as the function of time on stream.

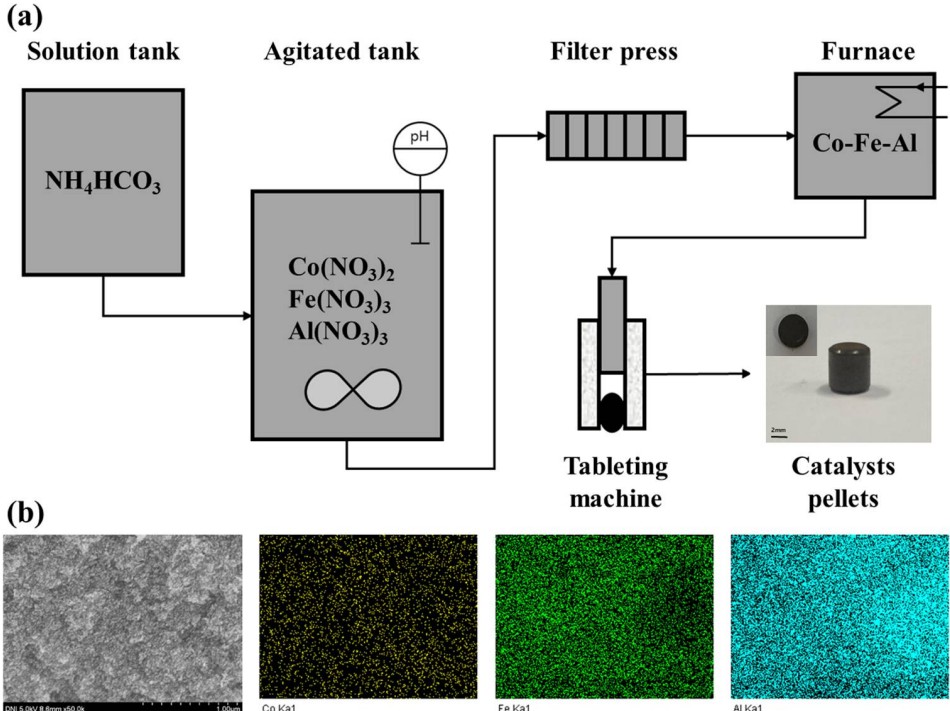

**Figure 5.** (**a**) Scheme of the pilot-scale synthesis and (**b**) SEM image and SEM–EDS.

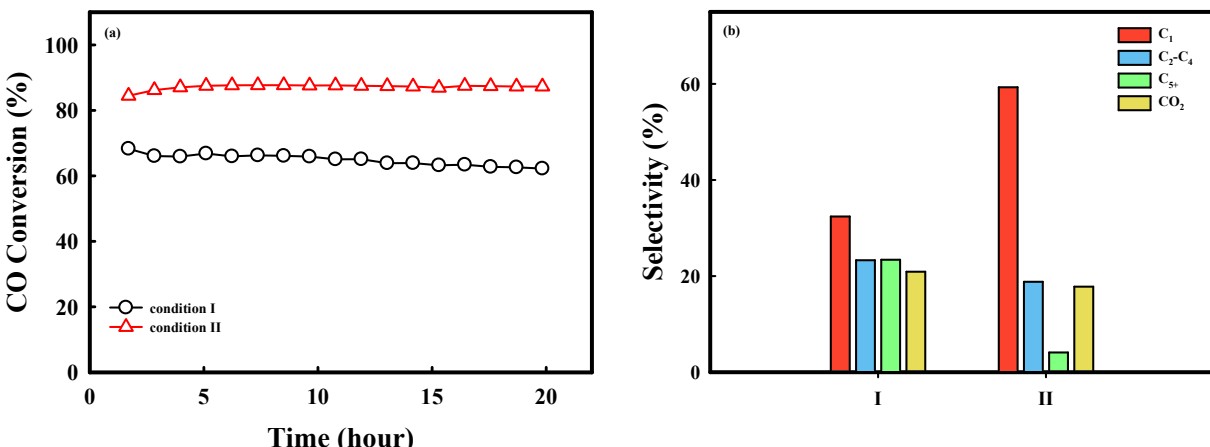

**Figure 6.** (**a**) The CO conversion with time on stream and (**b**) selectivity over the 40CFAl_P catalyst, under two different conditions ((I) 6000 mL/g/h, 300 °C; and 10 bar; (II) 4000 mL/g/h, 350 °C, and 20 bar).

Moreover, the results of initial hydrocarbon selectivity are shown in Figure 6b and summarized in Table 3. No remarkable change in selectivity was observed for 20 h (Figure S4). The CO conversion of 40CFAl_P increased dramatically under condition (II) compared to the case under condition (I). It is well known that elevated pressures, temperatures, and residence time improve the CO conversion [13]. As shown in Figure 6b, under condition (II), the selectivity for light hydrocarbons ($CH_4$ and $C_2$–$C_4$) increased up to 78.1%, whereas that for $C_{5+}$ and $CO_2$ decreased. It is known that light hydrocarbon contents ($C_1$–$C_4$) increase, whereas $C_{5+}$ hydrocarbon contents decrease at high temperatures under Fischer–Tropsch synthesis. In particular, the $CH_4$ selectivity dramatically increases; this is because the $CO_2$ methanation reaction is favorable at high reaction pressures and temperatures. In addition, under condition (II), the hydrocarbon time yield increased up to 1.56 $mmol_{CO} \cdot g_{metal}^{-1} \cdot h^{-1}$, despite the decrease in the $C_2$–$C_4$ hydrocarbon selectivity. Consequently, the 40CFAl_P catalyst affords a high CO conversion (87.6%), high light hydrocarbon selectivity ($CH_4$

59.3% and C$_2$–C$_4$ 18.8%), and low byproduct amounts (C$_{5+}$: 4.1% and CO$_2$: 17.8%) under condition (II).

**Table 3.** Summary of the catalytic performance over the 40CFAl_P catalyst under different conditions.

| Notation | Reaction Condition | CO Conversion (%) | Selectivity (%) | | | | C$_2$–C$_4$ Hydrocarbon Time Yield (mmol$_{CO}$·g$_{metal}$$^{-1}$·h$^{-1}$) |
| | | | CH$_4$ | C$_2$–C$_4$ | C$_{5+}$ | CO$_2$ | |
| --- | --- | --- | --- | --- | --- | --- | --- |
| 40CFAl_P | 6000 mL/g/h, 300 °C, 10 bar | 65.1 | 32.4 | 23.3 | 23.4 | 20.9 | 1.43 |
| | 4000 mL/g/h, 350 °C, 20 bar | 87.6 | 59.3 | 18.8 | 4.1 | 17.8 | 1.56 |

## 3. Materials and Methods

### 3.1. Catalysts Preparation

The CFAl catalysts were prepared at the laboratory scale (grams) by a coprecipitation method using mixed aqueous solutions of Co(NO$_3$)$_2$·6H$_2$O, Fe(NO$_3$)$_3$·9H$_2$O, and Al(NO$_3$)$_3$·9H$_2$O (Sigma-Aldrich, St. Louis, MO, USA), at room temperature. The total contents of Co and Fe in the catalysts were 20, 30, 40, and 50 wt.% (Co:Fe atomic ratio = 1:3). Next, aqueous ammonium bicarbonate was added dropwise to the mixed nitrate solution with stirring until pH = 7.0 ± 0.1 was achieved. Subsequently, the aged precipitate was filtered and washed several times with deionized water. The precipitate was dried at 110 °C for 12 h and subsequently calcined at 450 °C for 4 h. After calcination, the samples were sieved to remove catalyst particles smaller than 150 µm and larger than 250 µm. For convenience, the catalyst was denoted as xCFAl for Co–Fe–Al with different metal (Co and Fe) concentrations of x from 20 to 50 wt.%, with an Al$_2$O$_3$ support and a fixed Co/Fe ratio of 1/3. Additionally, the 40CFAl catalyst prepared at the pilot scale (kilograms) was obtained under conditions similar to those for the laboratory scale. The catalyst prepared at the pilot scale was pressed using an extruder to produce a pellet (5 × 5 mm$^2$), denoted as 40CFAl_P. The scheme of the pilot-scale synthesis is shown in Figure 5a.

### 3.2. Characterization

The metal content of the catalysts was measured with ICP-OES (PerkinElmer, Waltham, MA, USA). The textural properties of the catalysts were determined by N$_2$ adsorption–desorption isotherms, using a Micromeritics ASAP 2020 apparatus (Norcross, GA, USA) with N$_2$ isotherms at −196 °C. Prior to the adsorption measurements, the catalysts were purged at 200 °C for 5 h. The specific surface area was determined by the BET method. The XRD analysis was performed using a Phillips XPERT unit, with Cu–Kα radiation (λ = 1.5406 Å), at the Korea Basic Science Institute in Daegu. For the H$_2$-TPR analysis, the catalysts (0.2 mg) were heated from ambient temperature to 850 °C at a heating rate of 5 °C/min in 10 vol% H$_2$/N$_2$. The FT-IR spectra of the catalysts were recorded in the 4000–400 cm$^{-1}$ range, using a Spectrum GX & AutoImage spectrometer (PerkinElmer, Waltham, MA, USA). The samples were diluted with KBr.

### 3.3. Catalytic Test

The catalytic performance test was conducted in a fixed-bed stainless steel reactor (O.D. 1/2 in). In the experiment, 0.5 g of catalyst (150–250 µm) was loaded into a reactor. Prior to the catalytic test, the catalyst was pre-reduced in a 10 vol% H$_2$/N$_2$ gas mixture for 1 h at 500 °C and 100 mL/min. After the reduction, the catalysts were cooled to 200 °C under N$_2$ gas, after which the total gas flow (rate: 50 mL/min) was introduced into the reactor by mass-flow controllers at a space velocity (SV) of 6000 mL/g/h. The feed consisted of 72% H$_2$, 24% CO, and N$_2$ as the internal standard gas, which corresponded to a stoichiometric reactant ratio of H$_2$/CO = 3.0. The reactor pressure was increased up to 10 bar with syngas, using a back-pressure regulator. Next, the temperature of the reactor was increased from 200 °C to 300 °C at a heating rate of 10 °C/min to lessen the overshooting of the system temperature caused by the exothermic reaction. To avoid possible condensation of the

reaction products, the gas transfer lines were maintained at temperatures above 180 °C, and the heavy hydrocarbons were collected in a cold trap (4 °C) before analyzing the outlet gases online using a gas chromatograph (Agilent 6890, Santa Clara, CA, USA). The CO, $H_2$, $N_2$, and $CO_2$ gases were analyzed on a Carboxen 1000 column (Bellefonte, PA, USA) with a thermal conductivity detector (TCD, Santa Clara, CA, USA) and a GS-GASPRO capillary column (Agilent, Santa Clara, CA, USA) connected with a flame ionization detector (FID, Santa Clara, CA, USA) for analysis of the hydrocarbons. The CO conversion and selectivity for each product were calculated using Equations (1)–(3).

$$\text{CO conversion (carbon mole \%)} = \left(1 - \frac{\text{CO in the product gas (mol/min)}}{\text{CO in the feed gas (mol/min)}}\right) \times 100, \quad (1)$$

$$\begin{aligned}\text{Selectivity for hydrocarbons with carbon number n (carbon mole \%)} \\ = \frac{n \times C_n \text{ hydrocarbon in the product gas (mol/min)}}{(\text{total carbon} - \text{unreacted CO}) \text{ in the product gas (mol/min)}} \times 100,\end{aligned} \quad (2)$$

$$\begin{aligned}\text{Selectivity for carbon dioxide (carbon mole \%)} \\ = \frac{CO_2 \text{ in the product gas (mol/min)}}{(\text{total carbon} - \text{unreacted CO}) \text{ in the product gas (mol/min)}}.\end{aligned} \quad (3)$$

## 4. Conclusions

A set of CFAl catalysts with different metal loadings of 20, 30, 40, and 50 wt.% (Co/Fe ratio = 1/3) was prepared by the coprecipitation method and investigated for HC-SNG. The chemical structure of the components in the CFAl catalysts was strongly affected by the metal/alumina ratio, and the ratio of Co–Fe–Al spinel decreased with an increase in the metal content of the CFAl catalysts, thereby resulting in improved reducibility. Among the catalysts, 40CFAl was chosen as the optimum catalyst for the pilot-scale synthesis due to the best $C_2$–$C_4$ hydrocarbon time yield, and the catalyst was successfully reproduced in the pilot scale synthesis. Moreover, the pelletized catalyst (40CFAl_P) showed a high selectivity toward light hydrocarbons ($CH_4$: 59.3% and $C_2$–$C_4$: 18.8%) and low byproduct amounts ($C_{5+}$: 4.1% and $CO_2$: 17.8%) under the conditions of 4000 mL/g/h, 350 °C, and 20 bar.

**Supplementary Materials:** The following are available online at https://www.mdpi.com/2073-4344/11/1/105/s1, Figure S1: Pore size distribution curves over CFAl catalysts; Figure S2: CO conversion and hydrocarbon selectivity over CFAl catalysts (a–d) as a function of time on stream; Figure S3: CO conversion and hydrocarbon selectivity of the 40CFAl catalyst prepared at pilot scale as a function of time on steam; Figure S4: Time on stream of selectivity over the 40CFAl_P catalyst under two different conditions ((a) 6000 mL/g/h, 300 °C; and 10 bar; (b) 4000 mL/g/h, 350 °C, and 20 bar).

**Author Contributions:** T.Y.K. designed the experiments; S.C.L. and J.C.K. supervised the entire study; T.Y.K. performed the experiments and wrote the manuscript; S.B.J., J.H.W., J.H.L. and R.D. contributed to scientific discussions. All authors have read and agreed to the published version of the manuscript.

**Funding:** This work was supported by the Korea Institute of Energy Technology Evaluation and Planning (KETEP) and the Ministry of Trade, Industry and Energy (MOTIE) of the Korea (20203040030090).

**Institutional Review Board Statement:** Not applicable.

**Informed Consent Statement:** Not applicable.

**Data Availability Statement:** The data presented in this study are available on request from the corresponding author. Data is contained within the article or supplementary material.

**Acknowledgments:** This work was supported by the Korea Institute of Energy Technology Evaluation and Planning (KETEP) and the Ministry of Trade, Industry and Energy (MOTIE) of the Korea (20203040030090).

**Conflicts of Interest:** The authors declare no conflict of interest.

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
