# Peer review of "Investigation of Co–Fe–Al Catalysts for High-Calorific Synthetic Natural Gas Production: Pilot-Scale Synthesis of Catalysts"

_catalysts, doi:10.3390/catal11010105_

Round 1
Reviewer 1 Report
I have revised the paper which is very well written and easy to follow. The scientific part is done in a proper way and all the aspects are covered. It is interesting to see the difference when scaling up the catalyst synthesis process. This is always useful information for academia as well as industry. My suggestions for improvement are below:
Line 23, It is written: ‘’In addition, the CH4 distribution decreased, and the C2+ distribution increased as the metal loading increased.’’ I would rephrase to CH4 selectivity and C2+ selectivity instead of distribution. I suggest that authors use that terminology for the rest of the manuscript as well.
Line 41, It is written: ‘’Among transition metal catalysts, Ni is employed for methanation applications because of its high catalytic activity, high CH4 selectivity, and comparatively low cost.’’ The reference is missing
Line 42, It is written: ‘’Thus, Ni-based catalysts exhibit low C2+ selectivity.’’ I believe the word However, instead of Thus, would read much better.
Line 69, It is written: ‘’The textural properties of the CFAl catalysts with different metal loadings are listed in Table 1. The results of the Co and Fe loadings from ICP-OES analysis are also listed in Table 1.’’ It would better read in just one sentence, for example, The textural properties and ICP-OES…
Line 74, It is written: ‘’It has been shown that the catalysts with a high metal loading, prepared by co-precipitation, have a negative effect on the surface area [16,17].’’ It would be better for the paper if this statement is explained in the text with the reference.
Line 92, It is written: ‘’Conversely, with increasing the metal loading, the additional diffraction peaks of CoO, Fe3O4, and Fe0 were observed.’’ The authors should explain how does increasing the metal loading effect the catalyst particle size and crystallinity
Line 96, The text of Figure 1 is confusing, the word ‘’fresh’’ should be renamed to calcined
Line 143, It is written: ‘’With further 144 increasing the metal loading, the hydrocarbon time yield of 50CFAl was reduced to 2.24 mmolCO·gmetal−1·h−1 145 .’’ Why is that?
In table 2, why C5+ does not follow the trend? Why 20CFAL has the highest C5+ selectivity?
FigureS1 in the supplementary document is missing the legend (which symbol represents which hydrocarbon selectivity). Also, this figure should be mentioned in the text with an explanation.
Author Response
We would like to thank the reviewer for careful and thorough reading of this manuscript and for the thoughtful comments, which help to improve the quality of this manuscript.
Please, find the attached file.

Reviewer 2 Report
Comments to the manuscript (Catalysts 1045223) entitled: "Pilot- scale synthesis using co-precipitated Co-Fe-Al catalyst for high-calorific synthetic natural gas: effect of metal loading" by Tae Young Kim et al.
My general objection is that only one sample prepared in the bigger scale / pilot-scale (pellets 5x5 mm) was tested. The article title suggests that the effect of metal loading was evaluated for samples prepared in bigger scale. If 4 samples had been prepared in pilot- scale and one obtained in laboratory-scale had been chosen for comparison, it would have been understandable.
The same reactor and the same mass (0.5 g) was used in all catalytic tests (point 3.3). The differences observed in the activity of 40CFAl and 40CFAl_P were connected with mass transport limitations. It was not clearly explained which size of pilot-scale prepared catalyst particles was tested: 5x5 mm or the range: 150-250 μm, as for other samples? If the granulation was the same- what is the reason for lower activity obtained in the case of 40CFAl_P? This is, in my opinion, the most interesting question in this research.
Second objection: The use of H2 and N2 containing gas in conditions of high temperature, in the presence of Co-Fe catalysts causes a risk of ammonia production. That point was not taken into account in any experimental measurement reported in this work.
Another issue is connected with experimental part. According to Journal requirements: “The full experimental details must be provided so that the results can be reproduced.” It was not done here (form of the samples: powder, pellets, grains; purity of gases).
Generally: the chemical state of the catalyst components should be indicated for each sample, with increasing metal loading (metal, oxide, spinel?).
More specific comments are listed below:
1. Introduction: CO chemisorption was mentioned, in Experimental part: TG measurements. No first nor second was performed. Such situation cannot take place in the manuscript submitted to journal.
2. The trend obtained in BET results for the prepared series of catalysts should be explained. What is the reason for decreased S BET when moving from 40CFAl to 50CFAl?
3. XRD: the spinel phases should be clearly connected with the signals in XRD patterns (Fig. 1). The discussion of XRD results is incomplete: what is the reason for different intensity of peaks in Fig 1b? For example: signal connected with Fe (metallic) changes when moving from 40CFAl to 50CFAl?
4. TPR section: Why the reduction of Co and Fe occurs more easily at higher metal loadings? Is it connected with spinel connections, larger particle sizes or something else?
5. In point 2.2: Authors declare: "bimetallic Co-Fe catalysts exhibit a high selectivity toward the light hydrocarbons….compared to Co”. Suitable references would be helpful.
6. Point 2.2: The data presented in Fig. 4 are duplicated in Table 2. This should not take place in the manuscript. There is generally no discussion of the obtained activity results, even with the references cited earlier [18,19,20]. Beside 2 auto- citations [11,13], only one reference [22] was given in this part, but without deeper insight.
7. Point 2.3: the catalytic tests were conducted in 2 different conditions: I and II. Three parameters were changed simultaneously: gas flow rate (contact time was extended what generally causes the increase of conversion), temperature (it is commonly known that it also raises conversion; however one should take into account the thermodynamic equilibrium limitations) and pressure. I cannot understand how to connect the observed differences with only one parameter, for example temperature. It is impossible to make straight conclusions.
8. Point 2.3: Once again, the data presented in Fig. 6b are duplicated in Table 3. It is not clear, which point in time (from Fig 6a) was taken to present the data in Table 3: first point or after 10 hours or average? The obtained activity data should be discussed in respect of already published works from the same field. The most valuable would be papers from foreign research groups, if possible.
9. References should be formatted according to Journal instructions with careful checking of spelling. Some of them differ although the source is the same: [5,17].
10. Supplementary material: there is no legend on the chart. It is impossible to distinguish which line represents CO, methane etc.
Although the topic is really interesting, the lack of detail characterization of the samples, the lack of discussion of the activity results and the way of data presentation makes the manuscript incomplete and unclear.
Taking into account the above objections I do not recommend the manuscript to be published in Catalysts.
Author Response

(The authors gave the same response as above.)

Reviewer 3 Report
The topic of this manuscript is relevant to the scope of the Сatalysts. It deals with the Pilot-scale synthesis using co-precipitated Co–Fe–Al 2 catalysts for high-calorific synthetic natural gas. The objective of the present work is to study the catalytic performance with different metal (Co and Fe) loadings on a co-precipitated catalyst. .I think this manuscript may be published after minor revision as outlined below:
- It would be desirable to explain why specific surface area of CFAl catalysts (from 20 CFAl to 40 CFAl) increases and then (50 CFAl) decreases.
- I think it is necessary to add a small paragraph about stability of prepared catalysts. Will metal loading effect on the stability of CFAL catalysts.
Author Response

(The authors gave the same response as above.)

Round 2
Reviewer 2 Report
Review 2
Comments to the manuscript (Catalysts 1045223) entitled: " Investigation of Co–Fe–Al catalysts for pilot-scale in synthesis in high-calorific synthetic natural gas " by Tae Young Kim et al.
I really appreciate the effort that was made to improve the text. However, I still have some remarks. I will continue the convenient way of answering- proposed by Authors. It will be more clear and faster. My re- review comments , where present, will be given in italic font.
I. My general objection is that only one sample prepared in the bigger scale / pilot-scale (pellets 5x5 mm) was tested. The article title suggests that the effect of metal loading was evaluated for samples prepared in bigger scale. If 4 samples had been prepared in pilot- scale and one obtained in laboratory-scale had been chosen for comparison, it would have been understandable.
>>Thanks for your insightful comment. We investigated the effect of metal loading in Co-Fe-Al catalysts for high-calorific synthetic natural gas to find out the optimum composition, prepared the CFAl catalyst in pilot scale, and evaluate its catalytic performance. So, we revised the title as follows: “Investigation of Co–Fe–Al catalysts for pilot-scale synthesis in high-calorific synthetic natural gas”.
The new title proposed in second version is better but incomplete. I would suggest “Investigation of Co–Fe–Al catalysts for high-calorific synthetic natural gas production; pilot-scale synthesis of the catalyst”
II. The same reactor and the same mass (0.5 g) was used in all catalytic tests (point 3.3). The differences observed in the activity of 40CFAl and 40CFAl_P were connected with mass transport limitations. It was not clearly explained which size of pilot-scale prepared catalyst particles was tested: 5x5 mm or the range: 150-250 μm, as for other samples? If the granulation was the same- what is the reason for lower activity obtained in the case of 40CFAl_P? This is, in my opinion, the most interesting question in this research.
>>Thanks for your comment. In order to explain the activity of 40CFAl and 40CFAL_P more clearly, we modified the manuscript on lines 184-187 and 232-235 as follows:
“In addition, the effect of catalyst form was studied by changing powder to pellet, the catalytic performance of 40CFAl_P was investigated using pellet type under the same conditions as those for 40CFAl, to compare the CO conversion and selectivity”
“Additionally, the 40CFAl catalyst prepared at the pilot scale (kilograms) were obtained under conditions similar to those for the laboratory scale. The catalyst prepared at the pilot scale was pressed using an extruder to produce the pellet (5 x 5 mm), denoted as 40CFAl_P. The scheme of the pilot-scale synthesis is shown in Figure 5a”.
Thank you for the explanation. I understand that whole pellets were put into reactor. However, my question remains: what is the reason for lower activity obtained in the case of 40CFAl_P? Only activity test conditions (mass transport)? Would the activity be the same if the pellets will be crushed into 150-250 μm fraction? The suggestion what is the reason for the obtained results would really improve the practical impact of the manuscript.
III. Another issue is connected with experimental part. According to Journal requirements: “The full experimental details must be provided so that the results can be reproduced.” It was not done here (form of the samples: powder, pellets, grains; purity of gases).
>>Based on your comment, we correct experimental part.
“The CFAl catalysts were prepared at the laboratory-scale (grams) by a co-precipitation method using mixed aqueous solutions of Co(NO3)2·6H2O, Fe(NO3)3·9H2O, and Al(NO3)3·9H2O (Aldrich), at room temperature. The total contents of Co and Fe in the catalysts were 20, 30, 40, and 50 wt% (Co:Fe atomic ratio = 1:3). Next, aqueous ammonium bicarbonate was added dropwise to the mixed nitrate solution with stirring until pH = 7.0±0.1 was achieved. Subsequently, the aged precipitate was filtered and washed several times with deionized water. The precipitate was dried at 110 °C for 12 h and subsequently calcined at 450 °C for 4 h. After calcination, the samples were sieved to remove catalyst particles smaller than 150 μm and larger than 250 μm. For convenience, the catalyst was denoted as xCFAl for Co–Fe–Al with different metal (Co and Fe) concentrations of x from 20 to 50 wt.%, with an Al2O3 support and a fixed Co/Fe ratio of 1/3. Additionally, the 40CFAl catalyst prepared at the pilot scale (kilograms) were obtained under conditions similar to those for the laboratory scale. The catalyst prepared at the pilot scale was pressed using an extruder to produce the pellet (5 x 5 mm), denoted as 40CFAl_P. The scheme of the pilot-scale synthesis is shown in Figure 5a.”
I have carefully checked and the indicated fragment brings the same information that was given in lines 199-212 in the old version of the manuscript. I suggest to include the purity of gases used in the study.
IV. Generally: the chemical state of the catalyst components should be indicated for each sample, with increasing metal loading (metal, oxide, spinel?).
Unfortunately, that point was omitted by Authors in their reply. However, new data included in Table 1 are very helpful to understand what chemical compounds are present in four prepared samples: in calcined form and after reduction. My only remark is that spinel compounds: CoAl2O4 (01-082-2245 JCPDS) and FeAl2O4 were not clearly distinguished in Fig. 1 (only “spinel- like structure” is indicated) and Table 1.
I suggest that Authors should sum-up information obtained from XRD in one/ two sentences and extend the final text.
As to my specific comments:
2. The trend obtained in BET results for the prepared series of catalysts should be explained. What is the reason for decreased S BET when moving from 40CFAl to 50CFAl?
>> In order to explain the BET surface area with metal loading, we added reference and pore size distribution in Figure S1. So, we modified its description in line 72-77 of page 2 as follows:
“The BET surface area increased from 183 to 241.5 m2/g by increasing the Co and Fe loadings from 20 to 40 wt%, and further increase in the Co and Fe loadings caused a decrease in the BET surface area. The pore size distribution of CFAl catalyst are displayed in Figure S1, showing that the pore in 20CFAl catalysts almost have diameters between 10 and 40nm, while additional pore structure (4-8nm) was observed with increasing metal loading. It suggest that an increase of metal loading might increase the BET surface area due to formation of additional pore structure. However, for 50CFAl catalyst, the BET surface area decreased due to the crystallization of metal oxides (Co and Fe) at higher metal ratio [16- 18].”
“18. Li, F.; Li, X.; Liu, C.; Liu, T. Effect of alumina on photocatalytic activity of iron oxides for bisphenol a degradation. J. Hazard. Mater. 2007, 149, 199-207.”
The new Fig. S1 shows really interesting data. The statement that BET surface area increases due to formation of additional pore structure is rather obvious. The explanation for decrease of BET surface area as a consequence of crystallization of metal oxide particles is convincing. However, the Authors should discuss in more detail the opposite effect observed when moving from 20 to 40 wt.% metal loading. What is the reason of appearance of smaller pores in 30CFAl and 40CFAl in comparison to 20CFAl? The cited work [18] does not clarify that phenomenon.
4. TPR section: Why the reduction of Co and Fe occurs more easily at higher metal loadings? Is it connected with spinel connections, larger particle sizes or something else?
>> With increasing metal loading in CFAl catalysts showed low spinel phases, resulting in enhanced the reducibility as explained in the manuscript on page 4 lines 125-131 as follows:
“As shown in Figure 2b, the metal–Al spinel of the reduced catalysts rarely changed. Conversely, the adsorption bands of metal–oxygen decrease in intensity after the reduction at 500 °C. In particular, the 40CFAl catalyst showed a lower intensity compared to the 20CFAl catalyst. It was found that the reduction of the metal oxide occurs easily with increasing metal loading. Thus, it can be concluded that the low-ratio metal spinel is attributed to the improved reducibility with increasing metal loading, which is consistent with the XRD and TPR results.”
In addition, the calculated crystallite size of CoFeAl spinel phase in all of catalysts was similar. Thus, the reducibility might be affected by spinel connection than particle sizes.
I think there is a little mistake: “Figure 2b” should be written “Figure 3b” ?
Maybe I formulated my question not clearly enough and therefore I was misunderstood.
Authors reported that the low-ratio metal spinel was observed with increasing metal loading (XRD). They also shown that at higher metal loading improved reducibility was obtained (TPR). But this is only observation. I would like to know- what is the Authors suggestion about the reason for that trend? Is it because at low metal content in the sample whole amount of Co and Fe is in spinel form (not easy to reduce) and at higher metal loadings there is no Al enough to form more spinel so Co and Fe stay in oxide forms (easy to reduce)? If so, what is the reason that more Fe was observed in XRD for 40CFAl than for 50CFAl? H2-TPR data for 50CFAl suggest deeper reduction in that system compared to 40CFAl? It is really interesting issue. In my opinion any comment from Authors would really improve the discussion.
6. Point 2.2: The data presented in Fig. 4 are duplicated in Table 2. This should not take place in the manuscript. There is generally no discussion of the obtained activity results, even with the references cited earlier [18,19,20]. Beside 2 auto- citations [11,13], only one reference [22] was given in this part, but without deeper insight.
>>Thanks for your comment. In order to explain the activity result with the references more clearly, we revised its description in line 136-161 of page 5 as follows:
“The results of the catalytic behavior, such as the initial CO conversion and initial hydrocarbon selectivity as function of metal loading, are shown in Figure 4, and summarized in Table 2. There is no drastically change during the experimental, as shown in Fig. S2. With increasing the metal loading, the CO conversion increased remarkably as follows: 20CFAl (17.8%) < 30CFAl (54.4%) < 40CFAl (90.2%) < 50CFAl (97.0%). The fact that CO conversion increases dramatically with increasing metal loading may be due to the enhanced reducibility from low ratio of spinel phases at higher metal loading, compared to the 20CFAl catalyst. Above the
40CFAl catalyst, there was no significant change in the CO conversion. These experimental results correspond to the characteristic analysis including XRD, H2-TPR and FT-IR. Generally, bimetallic Co–Fe catalysts exhibit a high selectivity toward the light hydrocarbons (C2–C4) in the SNG (H2/CO = 3.0) compared to Co catalysts []. The CH4 selectivity decreased from 39.4% to 27.7%, the C2–C4 selectivity increased from 28.5% to 30.7%, and the C5+ selectivity increased from 7.6% to 16.4% with increasing metal loading, except for the 20CFAl catalyst. This is because the reducibility of Fe oxide phase increases with increasing metal loading, as mentioned above. It is well known that the CH4 selectivity of Fe catalysts decreases with increasing CO conversion at high pressures, due to the enhanced readsorption and reinsertion of olefins [11]. The CO2 selectivity is increase linearly with metal loading from 20 to 50wt.%. This increase in CO2 selectivity is caused by the (WGS) reaction (CO + H2O = CO2 + H2) at high H2O partial pressures; water vapor is produced in CO hydrogenation (paraffins: nCO + (2n+1)H2 = CnH2n+2 + nH2O, and olefins: nCO + 2nH2 = CnH2n + nH2O) [11,13,22]. This WGS reaction also leads to increase in H2/CO ratio, which affects the hydrocarbon selectivity in Fischer-Tropsch reaction [13]. In the case of 20CFAl catalyst, the H2/CO ratio adjustment is small due to the lower CO conversion, compared to other catalysts [13]. Thus, it is difficult to compare the hydrocarbon selectivity of 20CFAl with that of other catalysts. In addition, the paraffin ratio (P/(P+O)) was calculated with increasing metal amount, where P and O represent the paraffins and olefins in the C2-C4, respectively. The paraffin ratio also increased from 0.60 to 0.94 with increasing CO conversion. The high H2/CO ratio from WGS increase with increasing CO conversion, resulting in improvement of the paraffin ratio [].”
The new figure S2 is really interesting. The additional explanation gives deeper insight to the obtained trends. However, there is still no comparison between the obtained activity results and the similar catalysts presented in the literature, for example [18,19,20,22?].
10. Supplementary material: there is no legend on the chart. It is impossible to distinguish which line represents CO, methane etc.
>> Based on your comment, we added the legend in Figure S3.
Figure S3. The CO conversion and hydrocarbon selectivity of the 40CFAl catalyst prepared at pilot-scale as function of time on steam.
Thank you, the legend was necessary. However, I suggest careful checking of data series since their order in Fig. S3 is different than in old Fig. S1 (C2-C4: triangles higher than squares C5, or inversely?). The Authors should also add under which conditions (I or II) the data were obtained? CO conversion (above 80%) suggests cond. II, but Table 3 shows selectivity toward C5 around 4%- there is no such series on the chart.
Author Response
We would like to thank the reviewer for careful and thorough reading of this manuscript and for the thoughtful comments, which help to improve the quality of this manuscript.
Please, find the attached file
